# Human- or object-like? Cognitive anthropomorphism of humanoid robots

Alessandra Sacino[1], Francesca Cocchella[1,2‡], Giulia De Vita[1‡], Fabrizio Bracco[1‡], Francesco Rea[3‡], Alessandra Sciutti[2‡], Luca Andrighetto[1]*

1 Department of Educational Science, University of Genova, Genova, Italy, 2 Cognitive Architecture for Collaborative Technologies Unit, Istituto Italiano di Tecnologia, Genova, Italy, 3 Robotics Brain and Cognitive Sciences Unit, Istituto Italiano di Tecnologia, Genova, Italy

◉ These authors contributed equally to this work.
‡ FC, GDV, FB, FR and AS also contributed equally to this work.
* luca.andrighetto@unige.it

**Data Availability Statement:** The data underlying the results presented in the experiments are available on OSF at https://osf.io/fyp4x/.

**Funding:** This work has been supported by Curiosity Driven (2017)- D36C18001720005 grant

## Abstract

Across three experiments ($N = 302$), we explored whether people cognitively elaborate humanoid robots as human- or object-like. In doing so, we relied on the inversion paradigm, which is an experimental procedure extensively used by cognitive research to investigate the elaboration of social (vs. non-social) stimuli. Overall, mixed-model analyses revealed that full-bodies of humanoid robots were subjected to the inversion effect (body-inversion effect) and, thus, followed a configural processing similar to that activated for human beings. Such a pattern of finding emerged regardless of the similarity of the considered humanoid robots to human beings. That is, it occurred when considering bodies of humanoid robots with medium (Experiment 1), high and low (Experiment 2) levels of human likeness. Instead, Experiment 3 revealed that only faces of humanoid robots with high (vs. low) levels of human likeness were subjected to the inversion effects and, thus, cognitively anthropomorphized. Theoretical and practical implications of these findings for robotic and psychological research are discussed.

## Introduction

Robots are becoming more and more common in everyday life and accomplishing an ever-increasing variety of human roles. Further, their market is expected to expand soon, with more than 65 million robots sold a year by the end of 2025 [1]. As their importance for human life grows, the interest of robotics and psychology scholars in fully understanding how people perceive them constantly increases. Addressing this issue is indeed highly relevant, as one of the primary tasks of this technology is establishing meaningful relations with human beings.

The overall goal of the present research was to expand the knowledge about the human perception of robots. In doing so, we adopted an experimental psychological perspective on robotics (see [2]) and sought to uncover the cognitive roots underlying the anthropomorphism of these nonhuman agents.

to LA and funded by the University of Genova. The funders had no role in study design, data collection and analysis, decision to publish, or preparation of the manuscript.

**Competing interests:** The authors have declared that no competing interests exist.

## Anthropomorphizing robots

Research on Human-Robot interaction (HRI) provided convergent evidence that the appearance of robots, together with their behaviors [3, 4], deeply shapes people's perceptions and expectations. Basing on the design of robots, people form impressions on them and infer their peculiar qualities, such as likeability [5, 6], intelligence [7] or trustworthiness [5–9]. Although this design can assume different forms (e.g., machine- or animal-like), the humanoid shape is commonly considered as the most effective means to overcome the psychological barriers in the HRI [10]. Accordingly, humanoids are the robots most used within the social environment and, thus, the focus of the present research.

Similar to other nonhuman agents, the human likeness of robots is a key situational variable triggering people's tendency to anthropomorphize them [11]. That is, the perceived similarity of a humanoid robot to human beings increases people's accessibility to homocentric knowledge that is then projected onto the robot. Thus, robots resembling humans are more likely to be attributed distinctive human characteristics, such as the ability to think, being sociable [12], or feeling conscious emotions [13]. Further, such anthropomorphic inferences increase people's sense of familiarity with this nonhuman target and a sense of control over them, with subsequent benefits for the interaction [14]. A great deal of research corroborated this latter assumption, by for instance revealing that people tend to trust (e.g., [15]; see also [16]) or empathize [17] more with anthropomorphized robots, as well as expect that they can behave morally [18]. At the same time, the relationship between the perceived human likeness of robots and their acceptance in the social environment appears to be quite complex and not linear. Drawing from the Uncanny Valley hypothesis ([19], for a critical review see e.g., [20]), some researchers [21] have for example demonstrated that too high levels of anthropomorphic appearance of humanoid robots trigger a sense of threat towards them, as they are seen as undermining the uniqueness of human identity. In the same vein, robots perceived as too similar to humans are perceived as less trustworthy and empathic [9]. A humanoid appearance also implies the expectations that the robot should move and behave following human-like motion regularities. Such implicit belief, when not fulfilled (e.g., by a humanoid robot moving in a nonhuman like kinematics) hinders basic prosocial mechanisms such as automatic synchronization or motor resonance, reducing the possibilities to establish a smooth interaction [22]. In the same vein, perceiving this technology as too human-like heightens people's illusory expectations about the functions that this technology can indeed fulfill, and a violation of such expectations lowers the quality of HRI [23].

Despite the still debated effects of the human likeness of robots, anthropomorphism remains the most influential psychological process regulating the approach and subsequent interaction of humans with this technology. Thus, a systematic comprehension of the nature of this phenomenon is essential to better identify its antecedents and consequences for the HRI, be them positive or negative. So far, this process has been mostly conceived as a higher-order psychological process, consisting of inductive reasoning through which people attribute traits or qualities of human beings to this nonhuman agent. That is, most research in this field has investigated this process in terms of "content", by assessing the extent to which respondents are inclined to attribute uniquely human attributes (e.g., rationality or the capacity of feeling human emotions) to this technology.

Unlike these previous studies, the main purpose of this research is to examine this process through a "process-focused lens" [24], that is, investigating whether it could also occur at a more basic cognitive processing level. More specifically, we were interested in understanding whether people cognitively process humanoid robots as human- or object-like and whether the levels of human likeness endorsed by these robots may affect such cognitive processing.

Beyond contributing to the theoretical knowledge of this process, comprehending the cognitive roots of anthropomorphic perceptions could have important practical implications. How people cognitively perceive other agents (whether human or not) deeply shapes their first impressions—often at an unaware level—and affects the course also of HRI [25], above and beyond higher-order cognitive processes.

To achieve this aim, we integrated the existing research on the anthropomorphism of robots with cognitive paradigms commonly employed to study how people elaborate social (vs. non-social) stimuli.

## Configural processing of social stimuli and the inversion paradigm

During the last decades, cognitive psychology and neuroscience have intensively studied whether our brain processes social (e.g., a human face or body) and non-social stimuli (i.e., objects) similarly or differently. Cumulating evidence consistently reveals that people recognize social stimuli through configural processing, which requires considering both the constituent parts of the stimulus and the spatial relations among them. Such a process is activated both when people elaborate on human bodies (see [26] for a review) and faces (see e.g., [27] for a review). Instead, people recognize objects (e.g., a house) through analytic processing, which relies only on the appraisal of specific parts (e.g., the door), without requiring info about the spatial relations among them. Although the nature of this dual process is largely debated (see e.g., the expertise hypothesis, [28]) and it is still not clear whether human faces and bodies are unconditionally processed in a configural way, there is general agreement that such social stimuli are commonly elaborated in this way. In contrast, objects are commonly processed analytically.

The major indicator of this bias has been studied through the inversion paradigm, in which participants are presented with a series of trials first showing a picture of a social stimulus or an object, either upright or upside down. Afterward, subjects are asked to recognize the picture they just saw within a pair including a distractor (mirror-image). The main assumption is that when people are presented with a stimulus in an upside-down (vs. upright) way, their ability to process it by relying on the spatial relations of its constituent features should be impaired. Thus, this inversion should undermine the recognition of social stimuli as they are processed in a configural way, whereas it should not affect (or affect less) the recognition of objects, as they are analytically processed. Several investigations that also employed EEG methods [29] have confirmed such premise, first considering human faces (face-inversion effect, [30, 31]) and then bodies (body-inversion effect; [32]) as social stimuli. More recently, social psychology researchers have adapted the body-inversion paradigm to investigate the cognitive roots of sexual objectification. This is a specific form of dehumanization implying the perception (and treatment) of women as mere objects useful to satisfy men's sexual desires [33, 34]. In particular, Bernard and colleagues [35] demonstrated that the inversion effect (IE) does not emerge when people are exposed to images of sexualized female—but not male—bodies that were similarly recognized when presented upright or inverted. Hence, these social stimuli do not activate a configural processing and are cognitively objectified. This first impressive evidence has been then debated and criticized by Schmidt and Kistemaker [36], who demonstrated that the body asymmetry of the (female) stimuli used by Bernard and colleagues [35] explained the emerged pattern of findings (for a detailed discussion of this issue see [37, 38]). However, subsequent studies (e.g., [39]) employing a different set of stimuli controlled for their asymmetry confirmed the effect found by Bernard and colleagues [35], supporting the idea that the IE is a valid indicator to study the cognitive objectification of sexualized women [40].

Drawing on these studies, in the present research we adapted inversion paradigms as basic tools to systematically investigate an inverse process rather than objectification, people's

perception of nonhuman agents (i.e., robots) as human ones. Interestingly, Zlotowski and Bartneck [41] found preliminary evidence about the investigated process. Although not systematically checking for the stimuli asymmetry, they showed that robot images, similar to human ones, were subjected to the IE and thus processed in a configural way. The main goal of the present research is replicating and expanding this initial evidence in different ways. In the first step, we aimed to verify whether the IE would emerge for robot stimuli when controlling for each employed stimulus's asymmetry. Second, we verified whether the human-like appearance of humanoid robots would modulate the hypothesized cognitive anthropomorphism, and especially emerge for humanoid robots with high levels—but not with low levels—of human-like appearance. Third, we explored whether similar effects would emerge not only when considering the whole silhouettes of robots (body-IE), but also their faces (face-IE). In fact, we reasoned that an exhaustive comprehension of the cognitive anthropomorphism of humanoid robots should also encompass how human beings process their faces, besides their bodies. Faces are indeed the focal point in social cognition [42] and a prominent cue of humanity. Accordingly, recent research [43] for example revealed that (human) faces follow a peculiar configural processing, which in turn activates human-related concepts.

## Research overview

We designed three experiments to address the aims outlined above. In all the studies, we relied on inversion paradigms adapted from the previous studies, in which participants were exposed to stimuli portraying human beings, humanoid robots or objects. Following the original protocols, the image was first presented in an upright or inverted position for each trial and then followed by two images. One of them was the original picture and the second was its mirrored version (i.e., distractor). Participants' task was to recognize which picture of the two was the initial one.

In Experiment 1 and 2, participants were displayed entire bodies of human beings or humanoid robots, to investigate whether the body-IE would emerge both for human and robot stimuli. In Experiment 3, we explored the face-IE for the target stimuli, by presenting participants pictures portraying faces of human or humanoid robots. Further, in Experiment 1 we kept constant the medium levels of human likeness of robots and faces. Instead, in Experiment 2 and 3 we manipulated them by selecting robots with high vs. low scores of overall (Experiment 2) or face human likeness (Experiment 3; for more details about the selection of these stimuli see below). To increase the consistency of the investigated effects, across the studies we also varied the object-control stimuli, including human-like objects (i.e., mannequins; Experiment 1), buildings (Experiment 2) or general domestic tools (Experiment 3).

Finally, in all the studies we verified whether the cognitive anthropomorphism detected through the IE would be associated with the higher-order anthropomorphism, that is with respondents' tendencies to attribute robots uniquely human qualities.

## Experimental material

The prototypes of robots were initially selected from the ABOT database (http://abotdatabase. info/; [44]). It is a large pool of real-world humanoid robots that allows researchers to select them depending on their human-like appearance on distinct dimensions, each ranging from 0 to 100. In selecting our stimuli for robots, we set the filters for the considered dimensions, depending on our purposes and the availability of humanoid robot prototypes for the given range. That is, in Experiment 1, we selected 20 prototypes of robots with a medium overall human likeness score (42–66). In Experiment 2, we filtered 10 robots with a low overall human likeness score (0–40) and 10 robots with a high overall human likeness score (60–100).

In Experiment 3, we filtered 12 robots having a low overall human likeness score (0–45) and low human-like face score (0–42), plus 12 robots having a high overall human likeness score (60–100) and high human-like face score (60–100). Further, in Experiment 2 and 3 the body-manipulators filter was also used, by selecting robots having body-manipulators above 50. This allowed us to exclude robots composed of a single body part (e.g., a cube with only one eye, a single arm without head or body) and, thus, to obtain a more homogenous and comparable set of robots across the experiments and conditions.

For all the experiments, images of the selected robots were then retrieved online and standardized as follows. Using the open-source software Krita, all the images were uniformed in grayscale and pasted onto a white background. In Experiments 1 and 2, images of full body robots in a standing position and head directed towards the camera were edited to depict them from head to knee and fitted in a 397×576 pixels image. In Experiment 3, images of full front faces of humanoid robots with a neutral expression were trimmed, to remove external features and depict them from the hairline to the neck and then fitted in a 300×400 pixels image. Examples of the standardized stimuli of robots used in each experiment are displayed in Fig 1.

Concerning human stimuli (see Fig 2 for some examples), for Experiment 1, we selected 20 images from work by Cogoni and colleagues [39]; personalized conditions), portraying the whole silhouette of 20 individuals (10 men and 10 women) wearing casual clothes. To increase the generalizability of the hypothesized effects, in Experiment 2 we ad hoc created a set of human stimuli, portraying the entire body of 10 individuals (5 men and 5 women), each in two different poses. Similarly, in Experiment 3 we used a set of human stimuli ad hoc developed, consisting of 12 pictures of full front human faces (6 men and 6 women) with a neutral expression. Human stimuli were standardized through the same procedure used for the robot ones.

As object-control condition (see Fig 3), in Experiment 1 20 mannequins (10 male and 10 female) images were considered and standardized in the same way we did with robots and humans. Instead, in Experiment 2 (20 images) and 3 (12 images), we considered images of buildings as the object category, retrieved by the Cogoni and colleagues [39] research. Finally,

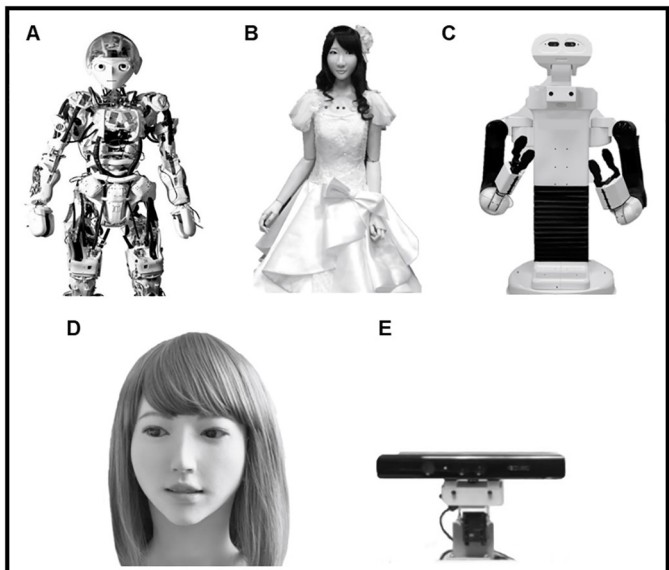

**Fig 1. Examples of humanoid robot stimuli employed in Experiment 1 (A), 2 (B, C), and 3 (D, E).**

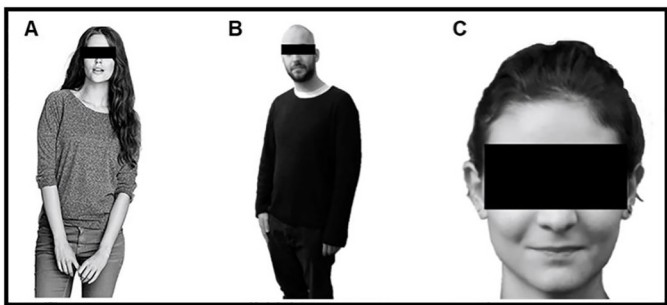

**Fig 2. Examples of human stimuli employed in Experiment 1 (A), 2 (B) and 3 (C).**

in Experiment 3, a new set of 12 object stimuli was created ad hoc, including a wide variety of domestic tools (e.g., a cup or a bottle).

Relevantly, for the experiments testing the body-IE (Experiment 1 and 2), an asymmetry-index was calculated for each robot, human and mannequin stimulus, following the procedure used in previous works [36–39]. For both experiments, data analyses revealed that the degree of asymmetry of the stimuli did not differ across the considered categories (see S1 File for more details about the procedure and data analyses).

## Open science practices and statistical methods

The sample sizes for all the experiments were a priori planned following the recommendation by Brysbaert [45], who suggested that around 100 participants are requested to have adequate power when focusing on within-subjects designs with repeated-measures variables and interactions between them. For each experiment, we reported all the stimuli, variables, and manipulations. All data and materials are posted and publicly available on OSF at https://osf.io/fyp4x/.

Main analyses were conducted using the GAMLj package [46] in Jamovi 1.8.4 version (The Jamovi project, [47], using a generalized mixed-model with a logit link function (logit mixed-model; [48]). In all the experiments, we considered participants' binary accuracy responses as the main outcome variable, coded as correct (1) and incorrect (0). Also, as in each experiment, all the participants were presented the same set of stimuli, in our models we included both a by-subject and a by-item random intercept to account for individual variability and non-independence of observation. Stimulus orientation (upright = 1 vs. inverted = 2) and category

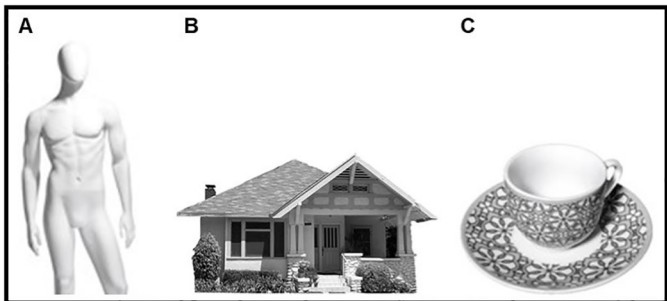

**Fig 3. Examples of object-control stimuli employed in Experiment 1 (A), 2 and 3 (C).**

(human vs. robot vs. control) were instead included as fixed effects. We reported significant odds ratios (OR) and the related 95% CI in interpreting the participants' accuracy. As our logit mixed-models predicted the odds of giving a correct response (accuracy = 1), a significant OR below 1 indicates that changes in the independent variable (e.g., presenting an image in the inverted orientation vs. the upright one) reduce the odds of giving a correct response, while a significant OR greater than 1 indicates an increase in the odds of giving a correct response.

Finally, in each experiment before running the main analyses, we performed an outlier analysis on the latency responses, based on the nature of our studies and the statistical mixed-model approach adopted [49, 50]. That is, we did not consider participants' responses on trials with latencies deviating more than ± 3 *SD* from the mean or with latencies below 50 *ms* (for a similar procedure, see [32]).

## Experiment 1

The first experiment was mainly designed to have preliminary evidence about the cognitive anthropomorphism of humanoid robots, relying on the body-IE. That is, we verified whether images portraying full-bodies of humanoid robots with medium levels of overall human likeness score would be cognitively elaborated similar to those of human beings and, thus, better recognized when presented upright than inverted.

### Method

**Ethics.** Procedures performed in both experiments were approved by the Departmental Ethics Committee (CER-DISFOR) and were in accordance with the APA ethical guidelines, the 1964 Helsinki Declaration and its later amendments. Written informed consent was obtained before participants started the experiments, and they were fully debriefed after each experimental session.

**Participants and experimental design.** Ninety-nine undergraduates at a north-western Italian university (39 male; $M_{age}$ = 22.2; $SD$ = 2.26) were recruited on a voluntary basis by research assistants via e-mail or private message on social networks. A snowball sampling strategy was used, with the initial participants recruited through the experimenters' friendship networks. A 2 (stimulus orientation: upright vs. inverted) × 3 (stimulus category: humans vs. robots vs. mannequins) within-subject design was employed.

**Procedure.** Participants came into the laboratory individually for a study "investigating the social perception towards human and nonhuman stimuli". The recognition task was administered using PsychoPy v3.03. Each participant was presented with 60 experimental stimuli (20 for each category) that were presented in a randomized order. Half of them were presented in an upright orientation and the other half 180˚ rotated on the *x*-axis (inverted condition). Following previous inversion-effect protocols, each trial began with the original image presented for 250 *ms* at the center of the screen in an upright or inverted orientation, depending on the experimental condition. Following a transient blank screen (1000 *ms*), participants were presented with two images, on the right and left of the center of the monitor, respectively. One image was the original one, the other was its mirrored version. Participants' task was to detect which of the two images was the same as the original one, by pressing the "A" key on the keyboard if the target image appeared on the left, the "L" key if it appeared on the right. Once participants had provided their responses, the next trial followed (see Fig 4 for a trial example). Before the experimental trials, participants were familiarized with the task through 9 practice trials.

After the recognition task, the higher-order anthropomorphism of robots was detected by adapting the 7-item (α = .82; $M$ = 1.55; $SD$ = 0.57) self-report scale by Waytz and colleagues

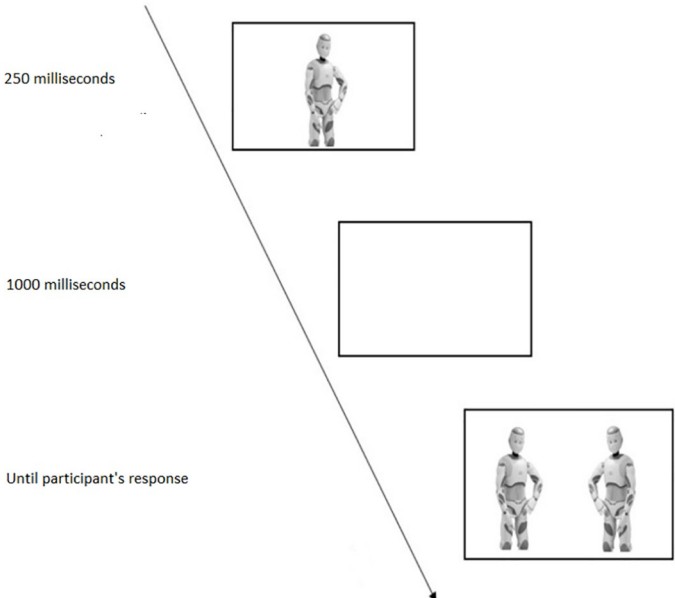

**Fig 4. A schematic representation of an experimental trial for the upright condition and humanoid robot as stimulus category.**

[51]. That is, participants were asked to rate the extent to which (*1 = not at all*; *5 = very much*) they believed that the considered prototypes of robots were able to have a series of human mental abilities, such as "a mind of its own" or "consciousness".

## Results

The outlier analysis on the latency responses identified 55 trials (out of a total of 5940) deviating more than ± 3 *SD* from the mean or with latencies below 50 *ms* and were thus removed from the main analyses.

The logit mixed-model conducted on participants' accuracy responses (1 = correct; 0 = incorrect) revealed a main effect of the stimulus orientation (1 = upright; 2 = inverted), $\chi2$ (1) = 74.72, $p < .001$, OR = 0.57, 95% CI [0.50, 0.65], suggesting that presenting the stimuli in an inverted orientation reduces the odds of giving a correct response. Put differently, overall, the stimuli were better recognized when presented upright (estimated accuracy, EA = .83 ± .03) than inverted (EA = .74 ± .03). Further, a simple slope analysis (see Fig 5) revealed that human stimuli were recognized better when presented in an upright (EA = .82 ± .04) than inverted orientation (EA = .73 ± .05), $\chi2$ (1) = 23.70, $p < .001$, OR = 0.58, 95% CI [0.47, 0.72]. Most interestingly, a similar pattern also emerged for robot images, that were better recognized when presented in an upright orientation (EA = .83 ± .04) than an inverted one (EA = .75 ± .05), $\chi2$ (1) = 18.30, $p < .001$, OR = 0.62, 95% CI [0.49, 0.77]. A similar pattern was also observed for the mannequins, with a better performance when stimuli were presented upright than inverted (EA for upright vs. inverted = .85 ± .03 vs. .74 ± .05), $\chi2$ (1) = 34.00, $p < .001$, OR = 0.51, 95% CI [0.41, 0.64]).

Instead, neither the main effect of stimulus category ($\chi2$ (2) = 0.81, $p = .666$), nor the interaction Stimulus orientation × Stimulus category emerged as significant, $\chi2$ (2) = 1.43, $p = .490$.

Finally, we tested the relationship between the magnitude of the IE for robots and the composite score of the self-report scale assessing the respondents' higher-order

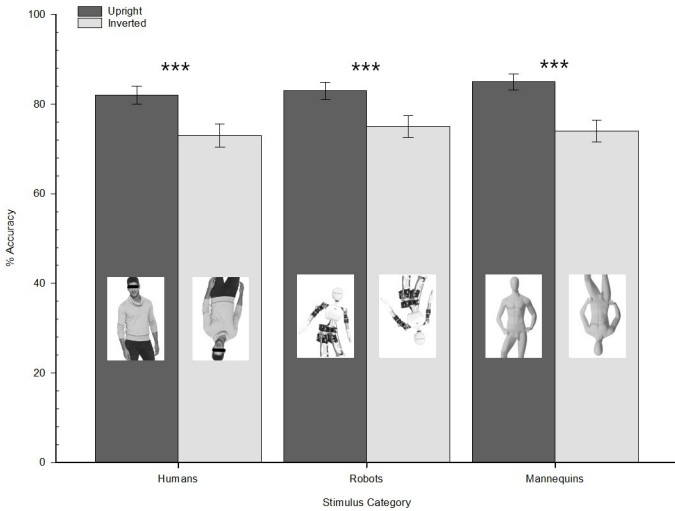

**Fig 5. Participants' estimated accuracy as a function of stimulus orientation (upright vs. inverted) and stimulus category (humans vs. robots vs. mannequins).** Experiment 1. Error bars represent standard errors of the mean values.

anthropomorphism. The IE index was obtained by subtracting for each respondent the accuracy mean of trials with robots in the inverted orientation from that of trials with robots in the upright orientation, so that the higher the value, the higher the magnitude of the IE. The correlational analyses revealed no significant link between the IE index and the respondents' higher-order anthropomorphism, $r = 0.04$, $p = 0.685$.

## Discussion

Findings from Experiment 1 provided initial evidence about the cognitive anthropomorphism of robots. By replicating the preliminary work by Zlotowski and colleagues [9] with a more controlled set of stimuli, we found that body images of humanoid robots with medium levels of human-like appearance were better recognized when presented in an upright than an inverted orientation. Thus, full body images of robots activated a configural processing, similarly to social stimuli portraying human beings. However, similar to previous work (see [39]), such body-IE also emerged for other objects with a human-body shape, i.e. mannequins. Thus, the question arises whether the human-like appearance of a given non-social stimulus triggers a configural processing per se, or whether the activation of the configural processing depends on the specific non-social stimulus considered. To address this issue, in Experiment 2 we manipulated the levels (high vs. low) of human-like appearance of full body images of robots, to verify whether the IE would be moderated by their degree of human likeness. Further, in Experiment 2 we employed a different set of stimuli than mannequins as the object-control condition. In particular, we used a pre-tested set of images portraying buildings, as these are a kind of object extensively used in previous research when exploring the IE of social vs. non-social stimuli.

Finally, unlike the previous study by Zlotowski and colleagues [9], in Experiment 1 we did not find evidence about a possible association between the cognitive anthropomorphism of robots (i.e., the magnitude of the IE for stimuli of robots) and the participants' higher-order anthropomorphism, which was detected in terms of attributions of uniquely human features. Thus, Experiment 2 was also designed to better investigate such relation.

## Experiment 2

### Method

**Participants and experimental design.** Ninety-four undergraduates at a north-western Italian university (40 male; $M_{age}$ = 21.8; $SD$ = 2.82) were recruited through a similar recruitment procedure to Experiment 1. In this experiment, a 2 (stimulus orientation: upright vs. inverted) × 4 (stimulus category: humans vs. robots with high human likeness vs. robots with low human likeness vs. buildings) within-subject design was employed.

**Procedure.** As the data collection for this and subsequent experiments took place during the COVID-19 pandemic, the recognition task was administered online using Inquisit 6 Web software. However, to ensure adequate control about participants' attention during the task, they were examined individually under the experimenter's supervision. She introduced them to the task and remained connected until the conclusion. Participants were then fully debriefed.

Each participant was presented with 80 experimental stimuli (20 per category). Unlike Experiment 1, all the stimuli were presented both in the upright and inverted orientation. This resulted in a total of 160 experimental trials per participant, preceded by 12 practice trials that helped familiarize themselves with the task. Due to the length of the task, the experiment was organized into four different blocks, each one containing 40 experimental trials and regarding a specific stimulus category. Stimuli were presented in a randomized order within each block, and the order of blocks was also randomized. Notably, before each block, participants were informed about the specific stimulus category that was presented. This information was especially important when considering the humanoid robots with high levels of human likeness, that would be per se not distinguished by human stimuli. The trial structure was similar to Experiment 1, presenting the original image (250 *ms*) followed by a blank screen (1000 *ms*) and the discrimination task.

After that, respondents' higher-order anthropomorphism of humanoid robots was detected by employing the same 7-item measure used in Experiment 1. In this experiment, participants were presented this measure twice in a randomized order, one referring to the robots with high human likeness ($\alpha$ = .87; $M$ = 1.59; $SD$ = 0.69), one referring to those with low human likeness ($\alpha$ = .82; $M$ = 1.47; $SD$ = 0.55). For each scale presentation, the target robots were shown at the top of the screen page.

### Results

The analysis on the latency responses identified 133 outlier trials (out of a total of 15040), that were thus removed from the main analyses.

The logit mixed-model conducted on participants' accuracy responses (1 = correct; 0 = incorrect) revealed a main effect of the stimulus orientation (1 = upright; 2 = inverted), $\chi2$ (1) = 84.18, $p$ < .001, OR = 0.66, 95% CI [0.60, 0.72]: overall, the stimuli were better recognized when presented upright (EA = .87 ± .02) than inverted (EA = .82 ± .03). Conversely, the main effect of stimulus category was not significant, $\chi2$ (3) = 0.66, $p$ = 0.883. Most importantly, the two-way Stimulus orientation × Stimulus category interaction emerged as significant, $\chi2$ (3) = 14.04, $p$ = .003. The interpretation of this interaction through the simple slope analyses (see Fig 6) revealed that robots with high levels of human likeness were more accurately recognized when presented upright (EA = .89 ± 0.3) than inverted (EA = .81 ± 0.5), $\chi2$ (1) = 46.95, $p$ < .001, OR = 0.53, 95% CI [0.44, 0.63]. Interestingly, a similar IE pattern also emerged for robots with low levels of human likeness (for upright orientation, EA = .88 ± 0.3; for inverted orientation, EA = .83 ± 0.5), $\chi2$ (1) = 20.43, $p$ < .001, OR = 0.66, 95% CI [0.55, 0.79]. Consistent with

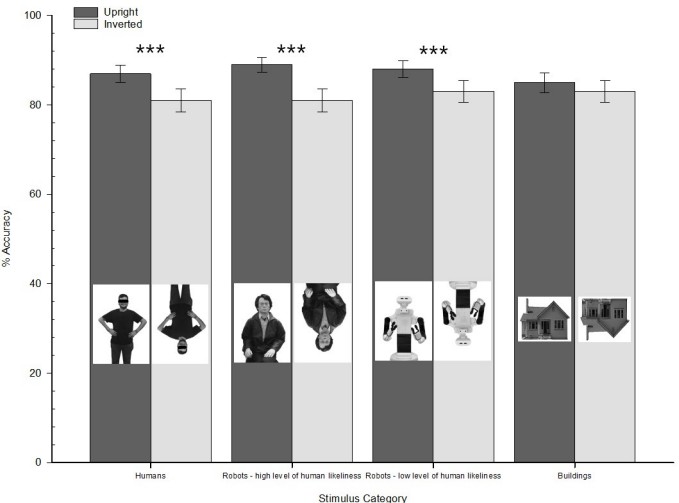

**Fig 6. Participants' estimated accuracy as a function of stimulus orientation (upright vs. inverted) and stimulus category (humans vs. robots with high human likeness vs. robots with low human likeness vs. buildings).**
Experiment 2. Error bars represent standard errors of the mean values.

Experiment 1, human stimuli were better identified when presented upright (EA = .87 ± 0.4) than inverted (EA = .81 ± 0.5), $\chi2$ (1) = 24.84, $p$ < .001, OR = 0.64, 95% CI [0.53, 0.76]. Instead, confirming previous literature, this pattern did not emerge as significant for buildings ($\chi2$ (1) = 3.49, $p$ = .062), indicating that participants had a similar performance in recognizing building stimuli regardless of their upright (EA = .85 ± 0.4) or inverted (EA = .83 ± 0.4) orientation.

Then, we verified the possible relation between participants' higher-order anthropomorphism of robots assessed through the self-report scale and their IE index, which was calculated similarly to the previous experiment. As the IE indexes for robots with high and low levels of human likeness did not differ ($t(93)$ = 1.55, $p$ = .124, 95% CI [-0.006, 0.053]), we collapsed them into a single one which was correlated with the composite scores of the self-report measures. In this case, also, the magnitude of the IE detecting the cognitive anthropomorphism did not correlate with the higher-order one, $r$ = -0.18, $p$ = .088.

## Discussion

The findings above replicated Experiment 1: once again, they revealed that the body-IE emerges for robots, similar to human beings. By expanding the previous results, the simple slope analyses also revealed that the body-IE was significant—and with a similar magnitude—when considering bodies of robots both with high and low levels of human likeness. Instead, consistent with previous literature, this effect did not emerge for objects (buildings).

Taken together, these results suggest that when cognitively processing full bodies of robot stimuli, perceivers tend to adopt a configural processing that is commonly activated for social stimuli. This process seems to regulate the cognitive elaboration of humanoid robots regardless of their levels of human likeness, at least when considering their full bodies. Consistent with the previous experiment, Experiment 2 revealed that this cognitive form of anthropomorphism is unrelated to the higher-order one: the IE index for robots did not significantly correlate with the self-report measure assessing the participants' tendencies to attribute human mental states to humanoid robots.

Experiment 3 was designed to expand these findings, by mainly verifying whether the IE also emerges when considering faces (i.e., face-IE) of humanoid robots, rather than full-bodies.

Like Experiment 2, we explored whether this presumed effect would be moderated by the levels (high vs. low) of human likeness of robot faces or, instead, emerge regardless of the degree of human likeness. Further, we compared the tested pattern of findings for robots with human facial stimuli and a series of object stimuli (i.e., domestic tools) created ad hoc. We opted to employ a different set of control stimuli to, on the one hand, increase the generalizability of our findings and, on the other hand, to have object stimuli with a size and a shape more comparable with the crucial stimuli of robot and human faces. Finally, we correlated the face-IE index of robots with a different scale of higher-order anthropomorphism than that used in the previous experiments.

## Experiment 3

### Method

**Participants and experimental design.** One hundred and nine undergraduates (52 male; $M_{age}$ = 22.1; $SD$ = 2.92) were recruited with a similar procedure used in the previous experiments. A 2 (stimulus orientation: upright vs. inverted) × 4 (stimulus category: human faces vs. robot faces with high human likeness vs. robot faces with low human likeness vs. objects) within-subject design was employed.

**Procedure.** Data collection was administered online using Inquisit 6 Web, following the same procedure employed in Experiment 2. Each participant was presented with 48 experimental stimuli (12 per category), presented in both upright and inverted orientation. This resulted in a total of 96 experimental trials per participant, preceded by 12 practice trials, that helped participants familiarize themselves with the task. Similar to Experiment 2, experimental trials were organized in 4 blocks, each one containing 24 trials, all regarding a specific stimulus category. Stimuli were presented in a randomized order within each block, the order of blocks was also randomized, and each block was followed by a pause. The trial structure was the same employed in Experiment 1 and 2, with the original image (250 ms) presentation followed by a blank screen (1000 ms) and the discrimination task.

After the computer task, respondents' higher-order anthropomorphism was measured. Unlike previous experiments, we employed an adapted version of the 4-item scale by Waytz et al. [52], which detected the extent to which (0 = *not at all*; 10 = *very much*) participants perceived the considered robots intelligent, able to feel what was happening around them, to anticipate what was about to happen or to plan an action in an autonomous way. In this experiment also the self-report measure was presented twice, one referring to the robots with high human likeness (α = .85; $M$ = 5.19; $SD$ = 2.64), one to those with low human likeness (α = .79; $M$ = 3.94; $SD$ = 2.39). The employed faces for these robots were displayed at the top of the page screen.

### Results

The outlier analysis on the latency responses identified 56 outlier trials (out of a total of 10464), that were thus removed from the main analyses.

Consistent with previous experiments, the mixed-model revealed a main effect of the stimulus orientation, $\chi2$ (1) = 32.80, $p < .001$, OR = 0.74, 95% CI [0.66, 0.82]: overall, stimuli were better recognized when presented upright (EA = .85 ± .03) than inverted (EA = .81 ± .03). The main effect of stimulus category also emerged as significant ($\chi2$ (3) = 29.40, $p < .001$), as well as the two-way Stimulus orientation × Stimulus category interaction, $\chi2$ (3) = 19.80, $p < .001$. Specifically, a simple slope analysis (see Fig 7) revealed that the IE emerged for human faces ($\chi2$ (1) = 38.46, $p < .001$, OR = 0.53, 95% CI [0.43, 0.65], EA for upright vs. inverted = .86 ± .04 vs. .77 ± .05) and robot faces with high levels of human likeness ($\chi2$ (1) = 15.65, $p < .001$, OR = 0.67, 95% CI [0.54, 0.81], EA for upright vs. inverted = .85 ± .04 vs. .79 ± .05). Instead,

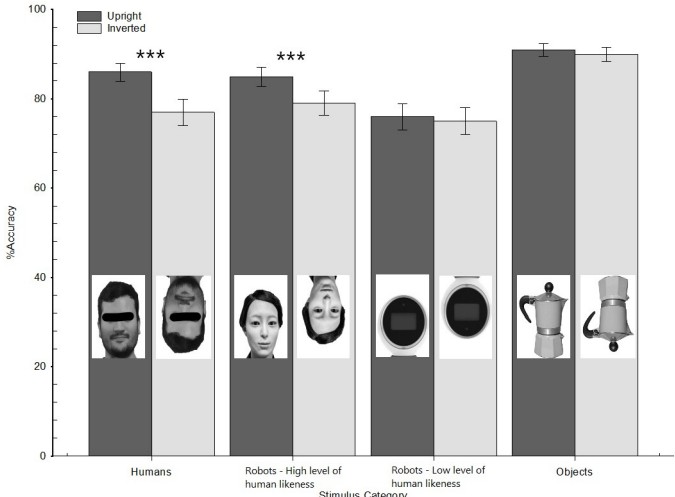

**Fig 7. Participants' estimated accuracy as a function of stimulus orientation (upright vs. inverted) and stimulus category (human faces vs. robot faces with high human likeness vs. robot faces with low human likeness vs. objects).** Experiment 3. Error bars represent standard errors of the mean values.

this pattern did not emerge as significant for robot faces with low levels of human likeness, $\chi2$ (1) = 0.68, $p$ = .411 (EA for upright vs. inverted = .76 ± 0.6 vs. .75 ± 0.6). Similarly, participants had a similar performance in recognizing objects regardless of their orientation, $\chi2$ (1) = 0.70, $p$ = .404 (EA for upright vs. inverted = .91 ± 0.3 vs. .90 ± 0.3).

Finally, we calculated the correlation between the higher-order anthropomorphism detected through the self-report measure and the face-IE index, which was calculated like previous experiments. As in this experiment the IE emerged for faces of humanoid robots with high levels but not for those with low levels of human likeness, we computed separated correlations. In that case, also, the relation between the IE index and the participants' higher-order tendencies to anthropomorphize robots was not significant, neither when considering the facial stimuli of robots with high levels ($r$ = − 0.06, $p$ = .552) nor when considering those with low levels of human likeness ($r$ = .04, $p$ = .639).

## Discussion

Findings for Experiment 3 revealed that the IE for robots also occurs when considering as stimuli their faces, rather than the entire bodies. Thus, the configural processing for this technology seems to be validated both for the body- and face-IE. However, the simple slope analyses conducted for this experiment revealed that the degree of human likeness of robots affects the face-IE. It emerged only for facial stimuli of humanoid robots with high levels of human likeness, but not for those with low levels. In line with previous experiments and literature, the IE also occurred for human facial stimuli but not for objects, also when employing a set of stimuli (i.e., domestic tools) different from Experiment 2. Finally, consistent with previous experiments the cognitive anthropomorphism of humanoid robots detected through the IE index did not correlate with the higher-order anthropomorphism assessed through the self-report measure.

## General discussion

Overall, findings from our three experiments provided convergent evidence about the human tendency to cognitively anthropomorphize humanoid robots. Similar to stimuli portraying

human beings, robots were consistently better recognized when presented in an upright than inverted orientation. Hence, they were subjected to the IE and processed in a configural way, like social stimuli. Instead, confirming previous literature (e.g., [53]), our results revealed that an analytic processing was triggered when participants visually processed a wide range of objects non-resembling human beings (i.e., buildings and domestic tools).

However, we found relevant differences when considering the full body of robots (body-IE, Experiment 1 and 2) and their faces (face-IE, Experiment 3). In fact, if the body-IE emerged for all levels of the human likeness of robots (medium, Experiment 1; low and high, Experiments 2), the face-IE only emerged for humanoid robots with high levels of human likeness, but not for those with low levels. We argue that such a different pattern of results may depend on the perceptual cues elicited by humanoid robots' full bodies or faces. More specifically, it is plausible to imagine that when considering the full bodies of robots, few anthropomorphic visual cues are necessary to trigger a configural processing, such as a single arm, a leg, or only the chest. This may explain why humanoid robots with low levels of human likeness are subjected to IE. Such assumption is also indirectly supported by Experiment 1 which considered the object-control category of mannequins. In fact, coherent with previous works [39], our findings revealed that these human-body-like objects are subjected to the IE and thus trigger a humanized representation at a cognitive level. Therefore, we may speculate that when considering the full body as a crucial stimulus, few visual features resembling human beings are sufficient to activate a configural processing, presumably above and beyond the semantic category within which each stimulus is classified (human being vs. object).

Conversely, when considering the faces of robots, the results of Experiment 3 suggest that a high level of human likeness is required to enact a configural processing. We believe that this is a highly relevant finding that highlights the prominent role of the face in defining the perceived full humanity (or no humanity) of a given exemplar, also at a cognitive level. That is, it is possible that, unlike the entire body, when focusing on the key component of the face, people need meaningful cues resembling human beings before activating a humanized representation of robots and a consequent configural processing. This argumentation is also in line with the work by DiSalvo and colleagues [54], which indicated that the faces of robots require the presence of specific and multiple features to be perceived as human-like (e.g., nose, eyelids, and mouth). These specific features can be observed in humanoid robots with high levels of human likeness (e.g., Erica and Sophia robots in our Experiment 3), whilst robots with low levels of human likeness often lack these features. For example, most robots with low levels of human likeness included in the ABOT database and thus employed in Experiment 3, despite having a head, did not show specific human features, as their head was made by a combination of object-like components (e.g., a monitor or a camera combined with a set of microphones). Instead, only a few of these robots (e.g., the Poppy robot) had eyes and eyebrows.

Taken together, we believe that our findings meaningfully extend research on the social perception of robots in several directions. First, we demonstrated that their anthropomorphic perceptions also have a cognitive basis, at least when considering humanoid robots. As mentioned when introducing our research, such an overall finding has great importance, because how people cognitively perceive robots deeply affects the first impressions toward them and the possible course of the HRI. That is, our results revealed that on the cognitive level humanoid robots can be elaborated not as mere objects but as social agents and, thus, they presumably trigger anthropomorphic knowledge and expectations, also at an unaware level. Such activation should primarily have positive outcomes for the HRI. In fact, most scholars in the field agree that the higher the—implicit or explicit—anthropomorphic perceptions of robots, the higher the positive feelings or attitudes that human beings display toward them. However, a possible side effect should be taken into account, especially in the light of our results revealing

that these anthropomorphic perceptions could be rooted in first-order cognitive processes. That is, similar to other technologies [55], heightened expectations that humanoid robots can be like human beings can increase negative emotions and attitudes toward them when such expectations are disregarded.

Second and besides that, for the first time in the literature, our results indicate that the cognitive anthropomorphic perceptions of humanoid robots may be different depending on the considered components of robots: while the body of humanoid robots triggers a humanized representation regardless of their levels of human likeness, their faces are cognitively perceived in anthropomorphic terms only when they highly resemble human beings. This latter finding could provide robotics engineering with relevant insights when planning and projecting the external features of robots. Further, our experiments importantly integrate and extend the preliminary evidence by Zlotowski and Bartneck [41]. In fact, they also revealed the occurrence of IE for robots, albeit considering a broader spectrum of full body (humanoid and no humanoid) robots that were not systematically checked and balanced for their asymmetry and human likeness. Unlike this single study, our experiments exclusively considered humanoid robots with different levels of human-like appearance (i.e., presence of body-manipulators) and thus may provide more specific indications about when (and if) these robots are cognitively recognized as human- vs. object-like. Further, across our experiments, we consistently did not find a linear relationship between the IE index for social robots and the people's explicit tendency to anthropomorphize them. This latter evidence contrasts Zlotowski and Bartneck [41]'s study, who instead found a positive linear relationship between the magnitude index of IE and the respondents' explicit tendency to attribute uniquely human traits and abilities to robots. These different results may be due to the different stimuli that Zlotowski and Bartneck [41] considered than our research, which encompassed a wider range of robots, including also non-humanoid ones. Such a wider spectrum may have triggered different anthropomorphic explicit tendencies than those elicited by humanoid robots that we considered across our experiments. Alternatively, unlike this previous study, our evidence may robustly confirm the idea that in social cognition implicit and first-order processes are often qualitatively different than those more conscious and elaborated (e.g., [56]) and may play a complementary or opposite role, depending on the considered social or no social target. Accordingly, implicit measures, such as the inversion effect paradigm that we employed in our research, commonly assess mental constructs (e.g., perceptions, attitudes) that are distinct from those detected through self-report measures. Put differently, implicit methods capture first-order cognitive processes that meaningfully contribute to explaining different aspects of social cognition, not accounted for by the corresponding explicit measures [57]. About this issue, we believe that our measure of cognitive anthropomorphism may capture one of the main psychological mechanisms underlying this phenomenon, i.e. the people's accessibility to anthropocentric knowledge [11], more appropriately than an explicit and self-report measure.

Despite the relevance of our findings, some limitations should be considered in interpreting them and driving the direction of future research. First, our experiments investigated the cognitive anthropomorphism of humanoid robots by relying only on the inversion effect paradigm. Although such a paradigm is the most extensively used when cognitively investigating the perception of social (vs. non-social) stimuli, we believe that future research should replicate our findings by employing further cognitive paradigms. For example, the whole vs. parts paradigm (see [58]) or the scrambled bodies and faces task (e.g., [53–59]) could be two further tools that could importantly strengthen the generalizability and robustness of our findings, by also better explaining the different cognitive elaboration of bodies and faces of social robots. With regard to this issue, it is also noteworthy that in our paradigm we explicitly differentiated the stimuli, both in the initial instructions and before each block. That is, participants were

made salient the stimulus category (i.e., human vs. robots) that they were going to be exposed to, and such a salience could have somewhat affected their cognitive elaboration. Thus, future research should investigate whether the pattern of findings that we found could be replicated also when the stimulus category is not made salient, especially when referring to robots with high levels of human likeness.

Second, our research only considered humanoid robots. We elected to focus on this specific type of robot for two main reasons. First, they are (and presumably will be) the most wide-spread prototypes of robots employed in social environments. Second, because focusing only on this type allowed us to obtain a more homogenous set of robots, which in turn made the comparison of the different levels of human likeness more reliable across the experiments and conditions. However, future research would compare the cognitive anthropomorphic perceptions of humanoid robots with those concerning object-like robots (e.g., Roomba), to verify whether only the first ones are indeed cognitively elaborated as social agents.

Third, similar to previous research on configural (vs. analytic) processing, we only considered images as experimental stimuli. Thus, future research would verify the cognitive anthropomorphism of robots by considering more ecologically valid stimuli or situations, that for instance could imply videos portraying robots or real brief interactions between participants and robots.

Fourth, in our experiments, we did not analyze whether people's levels of familiarity with humanoid robots would modulate their cognitive elaboration of these agents. More broadly, it would be interesting to verify across cultures possible differences in the cognitive anthropomorphism of robots, depending on people's habituation to living among humanoid robots. For instance, it is plausible to imagine that the cognitive anthropomorphism of robots would be especially high within the contexts in which these technologies are massively used in different domains of humans' everyday life.

## Conclusions

Robots are going to become an intrinsic component of our everyday life in a wide range of domains. Thus, a full comprehension of how people perceive and behave toward them is a primary task for psychology and engineering scholars. In achieving this purpose, we believe it is essential to integrate the knowledge about more explicit and conscious processes featuring the people's attitudes toward this technology with those concerning more cognitive processes underlying their perception. Both these processes play a pivotal and complementary role in understanding the factors facilitating or inhibiting the acceptance of robots in the social environment. In this sense, we hope that our research could provide important insights to think about and create robots as functional as possible in socially interacting with human beings.

## Supporting information

**S1 File.**
(DOCX)

## Author Contributions

**Conceptualization:** Giulia De Vita, Fabrizio Bracco, Francesco Rea, Alessandra Sciutti, Luca Andrighetto.

**Data curation:** Alessandra Sacino.

**Formal analysis:** Alessandra Sacino.

**Funding acquisition:** Luca Andrighetto.

**Investigation:** Francesca Cocchella, Giulia De Vita.

**Methodology:** Alessandra Sacino, Luca Andrighetto.

**Project administration:** Luca Andrighetto.

**Software:** Alessandra Sacino, Francesca Cocchella, Giulia De Vita.

**Writing – original draft:** Alessandra Sacino, Francesca Cocchella, Luca Andrighetto.

**Writing – review & editing:** Alessandra Sacino, Francesca Cocchella, Fabrizio Bracco, Francesco Rea, Alessandra Sciutti, Luca Andrighetto.

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
