## [Decision Letter · Decision Letter 0]

18 Mar 2022

PONE-D-21-30385Human- or object-like? Cognitive Anthropomorphism of Humanoid RobotsPLOS ONE

Dear Dr. Andrighetto,

Thank you for submitting your manuscript to PLOS ONE. After careful consideration, we feel that it has merit but does not fully meet PLOS ONE’s publication criteria as it currently stands. Therefore, we invite you to submit a revised version of the manuscript that addresses the points raised during the review process.

We look forward to receiving your revised manuscript.

Kind regards,

Josh Bongard

Academic Editor

PLOS ONE

Reviewers' comments:

Reviewer's Responses to Questions

**Comments to the Author**

1. Is the manuscript technically sound, and do the data support the conclusions?

Reviewer #1: Yes

Reviewer #2: Yes

2. Has the statistical analysis been performed appropriately and rigorously? 

Reviewer #1: Yes

Reviewer #2: Yes

3. Have the authors made all data underlying the findings in their manuscript fully available?

Reviewer #1: Yes

Reviewer #2: Yes

4. Is the manuscript presented in an intelligible fashion and written in standard English?

Reviewer #1: Yes

Reviewer #2: Yes

5. Review Comments to the Author

Reviewer #1: I have reviewed a manuscript entitled “Human- or object-like? Cognitive Anthropomorphism of Humanoid Robots”. This manuscript included three experimental studies well performed and with sufficient details (e.g., sharing materials, explaining the way they compute sample sizes). In general, I enjoy reading the manuscript. The main results are not always the ones authors were expected but they implement a rigorous procedure to test their hypothesis. Due to the strength of the methodology and the procedures, the authors are implementing I recommend this manuscript for publication in the journal.

See some comments after reading the manuscript:

Introduction: The introduction is well written and it covers the fundamental literature to understand the main goal of the manuscript. I would recommend authors to cover in a more detailed way the differences between processing faces (vs bodies) based on dehumanization literature so readers can understand the need to perform the third study while reading the introduction.

Method and results: Authors explain in a very detailed way the procedure to select the stimuli that they used in the experiments as well as report a priori sample size calculations and provide the link (osf) for the materials and manipulations. Study 1 was aimed to establish the effect of the cognitive anthropomorphism of humanoid robots using the inverted paradigm. The findings of this study are not the expected ones. I would like to see an attend of explaining why authors did not find the link between cognitive anthropomorphism of robots and the participants’ higher-order anthropomorphism when using the dimension of humanity they are measuring in this study. In studies 2 and 3 results are not always the expected ones. However, authors test their hypothesis by performing well-design studies, being well-powered, and with a rigorous procedure.

Discussion: In the discussion, section the authors acknowledged the limitations of their findings and discussed them in relation to previous evidence highlighting the understanding they have about the processes they are studying.

Reviewer #2: The current manuscript reports three studies that explored the human perception of robots. Overall, this is a very interesting topic, and I read this paper with enthusiasm. I believe that the present work could make a novel contribution to the literature.

There is much to like about the paper. It is well-written; it reports a set of three coherent studies which conceptually replicate one another. I consider that the general research goal is a highly valuable one, which may help understanding processes related to the way people anthropomorphize robots. The methodological approach to the phenomenon is also valuable. I thus think that the paper is of general interest for the readers of Plos ONE. However, I think the authors can improve the paper in some issues and minor details:

- First, I would strongly recommend review and reconsider the conclusions in several points. I wonder how the authors consider the contribution of the current research question to the study of human perception of robots. What are the consequences or expectancies of applying the same configural processing of social stimuli to robots? Is there any expected “side-effect”?

- Implications and alternative explanations for the weak correlation between the implicit measure and the explicit measure of anthropomorphizing found in the studies could be expanded. I think that you should include an extended comment in the discussion section about this point. Besides, you should include more theoretical explanations (and even consequences) about differences between those measures and cognitive processes behind them.

- What can we expect about human perception of robots in different cultures? Do you expect any effect depending on the familiarization with robots in daily life (e.g. people living in Tokyo or elsewhere, people working with robots? Is it relevant to consider the habituation to live among humanoid robots in their human perception? Please clarify and discuss to orient future research.

- Regarding the methodology, I wonder to what extent authors consider that the fact of explicitly differentiate the stimuli (“before each block, participants were informed about the specific stimulus category”) may affect the task. Is there any other way to proceed? Is this way different from the procedure used in previous research? Could this way to organise the procedure affect the explicit measure of anthropomorphism?

- To my view, the paragraph referring to sexual objectification in the discussion section is not properly justified, nor connected with the previous conclusions. Please reformulate.

I wish all these suggestions and minor critics could help to improve this outstanding line of research. Congratulations for this excellent work!

6. PLOS authors have the option to publish the peer review history of their article (what does this mean?). If published, this will include your full peer review and any attached files.

Reviewer #1: No

Reviewer #2: **Yes: **Naira Delgado

---

## [Author Response · Author response to Decision Letter 0]

4 Apr 2022

Dear Prof. Bongard,

On the 18th of March, we received your editorial decision regarding the manuscript (PONE-D-21-30385) entitled: “Human- or object-like? Cognitive Anthropomorphism of Humanoid Robots”.

We are very grateful to you and the Reviewers for the constructive comments to improve the paper and for allowing us to revise the manuscript and send a new version of it. 

In revising the paper, we took into consideration all the issues raised by you and the Reviewer, and we addressed them as follows (note that, for your convenience, changes in the manuscript are highlighted in yellow):

Editor – Issue 1: Please ensure that your manuscript meets PLOS ONE's style requirements, including those for file naming.

Reply: Before submitting the original and revised version, we carefully checked PLOS ONE's style requirements, by also referring to the online template.

Editor – Issue 2: Please include your full ethics statement in the ‘Methods section of your manuscript file. In your statement, please include the full name of the IRB or ethics committee who approved or waived your study, as well as whether or not you obtained informed written or verbal consent. If consent was waived for your study, please include this information in your statement as well.

Reply: Consistent with your suggestion, in this revised version we moved the ethics statement to the “Method” section of Experiment 1 (see p. 12, lines 259-263). Also, we specified the full name of the ethics committee and clarified that we obtained the informed written consent for each participant.

Editor – Issue 3: Please review your reference list to ensure that it is complete and correct. If you have cited papers that have been retracted, please include the rationale for doing so in the manuscript text, or remove these references and replace them with relevant current references. Any changes to the reference list should be mentioned in the rebuttal letter that accompanies your revised manuscript. If you need to cite a retracted article, indicate the article’s retracted status in the References list and also include a citation and full reference for the retraction notice.

Reply: We carefully reviewed our reference list both for the original and revised versions of the manuscript. We also mentioned in this letter the citations that we added in this letter (see below the replies to the issues and the reference list at the end of this letter). We did not include in our manuscript any retracted article.

Reviewer 1 – Issue 1: Introduction: The introduction is well written and it covers the fundamental literature to understand the main goal of the manuscript. I would recommend authors cover in a more detailed way the differences between processing faces (vs bodies) based on dehumanization literature so readers can understand the need to perform the third study while reading the introduction.

Reply: We would like to thank the Reviewer for this suggestion. Accordingly, in the introduction of this revised version (see p. 7, lines 154-158), we better highlighted the fact that, besides the bodies, exploring how people cognitively process and eventually anthropomorphize the faces of robots is crucial, as faces are a core point in social cognition [1] and their processing follows a peculiar path which activates human-related concepts [2].

Reviewer 1 – Issue 2: Method and results: Authors explain in a very detailed way the procedure to select the stimuli that they used in the experiments as well as report a priori sample size calculations and provide the link (osf) for the materials and manipulations. Study 1 was aimed to establish the effect of the cognitive anthropomorphism of humanoid robots using the inverted paradigm. The findings of this study are not the expected ones. I would like to see an attend of explaining why authors did not find the link between cognitive anthropomorphism of robots and the participants’ higher-order anthropomorphism when using the dimension of humanity they are measuring in this study. In studies 2 and 3 results are not always the expected ones. However, authors test their hypothesis by performing well-design studies, being well-powered, and with a rigorous procedure. 

Reply: We would like to thank the Reviewer for these thoughts. Indeed, we did not formulate specific hypotheses about the occurrence (or absence) of the correlation between cognitive and higher-order anthropomorphism of robots. In fact, if, on the one hand, a single piece of evidence [3] revealed a correlation between these two forms of anthropomorphism, on the other hand, a substantial body of social cognition literature revealed that implicit and first-order processes (e.g., the cognitive anthropomorphism of robots) are qualitatively different than more elaborated and higher-order processes (e.g., the anthropomorphism of robots when detected through self-report measures). Thus, as mentioned when introducing our experiments, we just explored the possible link between first- and higher-order anthropomorphism of robots, but we did not necessarily expect a significant correlation. Thus, the obtained results showing no significant correlation were not “unexpected”. 

Said that, by also following both Reviewers’ suggestions (see also the Issue 2 below), in this revised version (see p. 25, lines 571-578) we further expanded the implications and explanations about the absence of correlation between the index of cognitive anthropomorphism obtained through the IE and the score of explicit anthropomorphism.

Reviewer 2 – Issue 1: First, I would strongly recommend review and reconsider the conclusions in several points. I wonder how the authors consider the contribution of the current research question to the study of human perception of robots. What are the consequences or expectancies of applying the same configural processing of social stimuli to robots? Is there any expected “side-effect”?

Reply: We would like to thank the Reviewer for raising this issue. By also considering her issues below, the General Discussion of this revised version has been revised and further expanded. Regarding this specific issue, we better discussed the possible implications and consequences of applying the same configural processing of humans to robots, by also mentioning a possible side-effect (see pp. 23, 24, lines 533-545). That is, we reasoned that a specific rebound effect should be taken into account, especially in the light of our results revealing that these anthropomorphic perceptions could be rooted in first-order cognitive processes. That is, similar to other technologies (see e.g.,[4]), heightened expectations that humanoid robots can be as human beings can increase negative emotions and attitudes toward them when such expectations are disregarded.

Reviewer 2 – Issue 2: Implications and alternative explanations for the weak correlation between the implicit measure and the explicit measure of anthropomorphizing found in the studies could be expanded. I think that you should include an extended comment in the discussion section about this point. Besides, you should include more theoretical explanations (and even consequences) about differences between those measures and cognitive processes behind them.

Reply: We would like to thank the Reviewer for this suggestion. In the original version, we included a specific paragraph in the General Discussion in which we attempted to explain the reasons why the two measures of anthropomorphism did not correlate. That is, we first argued that in our experiments we considered only humanoid robots. Instead, the study by Zlotowski and Bartneck [3], which revealed a significant correlation between the IE index and the explicit measure of anthropomorphism, considered a wider spectrum of humanoid and non-humanoid robots that could have triggered different anthropomorphic explicit tendencies. Further, we stressed that the absence of correlation that emerged in our experiments robustly confirms the idea that in social cognition implicit and first-order processes are qualitatively different than the explicit ones, that are instead more conscious and elaborated. Following the Reviewer’s tip, in this revised version (see p. 25, lines 571-578) we expanded the theoretical explanations featuring the differences between first- and higher-order processes, together with their consequences. In doing so, we specifically referred to the theoretical works by Nosek and colleagues [5] and Epley and colleagues [6].

Reviewer 2 – Issue 3: What can we expect about human perception of robots in different cultures? Do you expect any effect depending on the familiarization with robots in daily life (e.g. people living in Tokyo or elsewhere, people working with robots? Is it relevant to consider the habituation to live among humanoid robots in their human perception? Please clarify and discuss to orient future research.

Reply: We would like to thank the Reviewer for this insight. We discussed this issue in the General Discussion section (pp. 26,27, lines 607-613). In particular, we argued that it would be interesting to verify across cultures possible differences in the cognitive anthropomorphism of robots, by assuming that the cognitive anthropomorphism of robots would be especially high within the contexts in which these technologies are massively used in different domains of humans’ everyday life.

Reviewer 2 – Issue 4: Regarding the methodology, I wonder to what extent authors consider that the fact of explicitly differentiate the stimuli (“before each block, participants were informed about the specific stimulus category”) may affect the task. Is there any other way to proceed? Is this way different from the procedure used in previous research? Could this way to organise the procedure affect the explicit measure of anthropomorphism?

Reply: We believe that the issue raised by the Reviewer is highly relevant. In planning our experiments, we followed similar procedures to those used in previous cognitive or socio-cognitive works. However, we acknowledge that informing participants about the stimulus category that they were going to see would make salient them the belonging category of the stimuli (i.e., humans vs. robots) and, thus, somewhat affect their cognitive elaboration. Thus, it would be interesting to verify whether similar effects also emerge when the specific stimulus category is not made salient, especially when referring to robots with high levels of anthropomorphism. Right now, we are running an experiment similar to the Exp 3 of this manuscript, but in which participants are just presented in a random order trial with humans or robots’ faces, without differentiating the stimuli both in the instructions and during the task. In this revised version, we discussed this interesting issue in the General Discussion (p. 26, lines 587-593).

Reviewer 2 – Issue 5: To my view, the paragraph referring to sexual objectification in the discussion section is not properly justified, nor connected with the previous conclusions. 

Reply: We agree with the Reviewer that the paragraph in the General Discussion referring to sexual objectification is rather disconnected from the other conclusions. For this reason, and also because the revised General Discussion has been substantially expanded than the previous version, we decided to cut this paragraph.

Summary

We greatly appreciate the opportunity to revise our manuscript. The comments made by the Editor and the reviewers were very helpful. We hope you will now agree that our manuscript is suitable for publication in Plos One. If more revisions are needed, however, we would be happy to make them.

References

1. Macrae C, Quadflieg S. Perceiving People. Handbook of Social Psychology. 2010

2. Hugenberg K, Young S, Rydell R, Almaraz S, Stanko K, See P et al. The Face of Humanity. Soc Psychol Personal Sci. 2016;7(2):167-175. doi:10.1177/1948550615609734.

3. Zlotowski J, Bartneck C. The inversion effect in HRI: Are robots perceived more like humans or objects? In Proceedings of 8th ACM/IEEE International Conference on Human-Robot Interaction (HRI). 2013. p.365-372. doi:10.1109/HRI.2013.6483611.

4. Crolic C, Thomaz F, Hadi R, Stephen A. Blame the Bot: Anthropomorphism and Anger in Customer–Chatbot Interactions. Journal of Marketing. 2021;86(1):132-148. doi: 10.1177/00222429211045687.

5. Nosek B, Hawkins C, Frazier R. Implicit social cognition: from measures to mechanisms. Trends Cogn. Sci. 2011;15(4):152-159. doi:10.1016/j.tics.2011.01.005.

6. Epley N, Waytz A, Cacioppo JT. On seeing human: A three-factor theory of anthropomorphism. Psychol Rev. 2007; 114(4): 864–886. doi:10.1037/0033-295X.114.4.864.

---

## [Decision Letter · Decision Letter 1]

21 Jun 2022

Human- or object-like? Cognitive Anthropomorphism of Humanoid Robots

PONE-D-21-30385R1

Dear Dr. Andrighetto,

We’re pleased to inform you that your manuscript has been judged scientifically suitable for publication and will be formally accepted for publication once it meets all outstanding technical requirements.

Kind regards,

Josh Bongard

Academic Editor

PLOS ONE

Additional Editor Comments (optional):

Reviewers' comments:

Reviewer's Responses to Questions

**Comments to the Author**

1. If the authors have adequately addressed your comments raised in a previous round of review and you feel that this manuscript is now acceptable for publication, you may indicate that here to bypass the “Comments to the Author” section, enter your conflict of interest statement in the “Confidential to Editor” section, and submit your "Accept" recommendation.

Reviewer #1: All comments have been addressed

Reviewer #2: All comments have been addressed

2. Is the manuscript technically sound, and do the data support the conclusions?

Reviewer #1: Yes

Reviewer #2: Yes

3. Has the statistical analysis been performed appropriately and rigorously? 

Reviewer #1: Yes

Reviewer #2: Yes

4. Have the authors made all data underlying the findings in their manuscript fully available?

Reviewer #1: Yes

Reviewer #2: Yes

5. Is the manuscript presented in an intelligible fashion and written in standard English?

Reviewer #1: Yes

Reviewer #2: Yes

6. Review Comments to the Author

Reviewer #1: Authors have correctly addressed my comments in the previos revision. I endorse publication of the present manuscritp.

Reviewer #2: (No Response)

7. PLOS authors have the option to publish the peer review history of their article (what does this mean?). If published, this will include your full peer review and any attached files.

Reviewer #1: No

Reviewer #2: **Yes: **Naira Degado

---

## [Editor Report · Acceptance letter]

7 Jul 2022

PONE-D-21-30385R1 

Human- or object-like? Cognitive Anthropomorphism of Humanoid Robots 

Dear Dr. Andrighetto:

I'm pleased to inform you that your manuscript has been deemed suitable for publication in PLOS ONE. Congratulations! Your manuscript is now with our production department. 

Kind regards, 

on behalf of

Dr. Josh Bongard 

Academic Editor

PLOS ONE